

# Malaria: biochemical, physiological, diagnostic, and therapeutic updates

Enas El Saftawy[1,2], Mohamed F. Farag[3], Hossam H. Gebreil[4], Mohamed Abdelfatah[3], Basma Emad Aboulhoda[5], Mansour Alghamdi[6,7], Emad A. Albadawi[8] and Marwa Ali Abd Elkhalek[4,9]

[1] Department of Medical Parasitology, Faculty of Medicine, Cairo University, Cairo, Giza, Egypt
[2] Department of Medical Parasitology, Armed Forces College of Medicine, Cairo, Egypt
[3] Department of Medical Physiology, Armed Forces College of Medicine, Cairo, Giza, Egypt
[4] Department of Medical Biochemistry & Molecular Biology, Armed Forces College of Medicine, Cairo, Egypt
[5] Department of Anatomy and Embryology, Faculty of Medicine, Cairo University, Cairo, Giza, Egypt
[6] Department of Anatomy, College of Medicine, King Khalid University, Abha, Saudi Arabia
[7] Genomics and Personalized Medicine Unit, College of Medicine, King Khalid University, Abha, Saudi Arabia
[8] Department of Anatomy, College of Medicine, Taibah University, Madinah, Saudi Arabia
[9] Medical Biochemistry & Molecular Biology, Faculty of Medicine, Ain Shams University, Cairo, Egypt

Corresponding author
Basma Emad Aboulhoda,
doctor.basma@hotmail.com

## ABSTRACT

**Background**. Malaria has been appraised as a significant vector-borne parasitic disease with grave morbidity and high-rate mortality. Several challenges have been confronting the efficient diagnosis and treatment of malaria.

**Method**. Google Scholar, PubMed, Web of Science, and the Egyptian Knowledge Bank (EKB) were all used to gather articles.

**Results**. Diverse biochemical and physiological indices can mirror complicated malaria *e.g.*, hypoglycemia, dyslipidemia, elevated renal and hepatic functions in addition to the lower antioxidant capacity that does not only destroy the parasite but also induces endothelial damage. Multiple trials have been conducted to improve recent points of care in malaria involving biosensors, lap on-chip, and microdevices technology. Regarding recent therapeutic trials, chemical falcipain inhibitors and plant extracts with anti-plasmodial activities are presented. Moreover, antimalaria nano-medicine and the emergence of nanocarrier (either active or passive) in drug transportation are promising. The combination therapeutic trials *e.g.*, amodiaquine + artemether + lumefantrine are presented to safely counterbalance the emerging drug resistance in addition to the Tafenoquine as a new anti-relapse therapy.

**Conclusion**. Recognizing the pathophysiology indices potentiate diagnosis of malaria. The new points of care can smartly manipulate the biochemical and hematological alterations for a more sensitive and specific diagnosis of malaria. Nano-medicine appeared promising. Chemical and plant extracts remain points of research.

# INTRODUCTION

In 2023, the World Health Organization (WHO) estimated cases infected with malaria by 249 million globally which seriously exceeded the pre-pandemic level reported in 2019

(∼233 million) by 16 million patients. The WHO now interrelates the surge in malaria reports with the continuous climate changes. The higher incidences of deaths occur in 61% of children under 5 years and 125 million pregnant females are at threat of malaria infections (*World Health Organization, 2023*). Severe malaria is a systemic parasitic disease with a high death rate. Human malaria is characterized by a complex life cycle with liver and blood stages. In *P. vivax* and *P. ovale* sporozoites convert into hypnozoites and remain latent in the hepatocytes for months or years causing relapses (*Nureye & Assefa, 2020*). Also, in *P. falciparum* and *P. malariae,* ineffective clearance of the parasite (recrudescent infection) or entire reinfection may frequently occur (*Plucinski et al., 2015*).

Both developing and developed nations are prone to the misdiagnose of malaria either due to scarcity of resources or poor acquaintance with the disease. Conventional microscopy of Giemsa stained thick and thin blood films despite being the standard method of diagnosis is time-consuming and demands proficiency; thus, various research managed to develop alternative diagnostic techniques. The main purpose is to improve sensitivity, and test duration, with competitive costs to allow efficient parasite identification and quantification (*Pham et al., 2018*). Malaria is strongly related to several biochemical and hematological changes. Yet, useful manipulation of these indices either in the routine laboratory panel or in the development of new "points of care" (*Liu, Ye & Cui, 2020*) has been introduced in the current review

Another issue is the challenges that face the long list of anti-malarial drugs involving quinine, primaquine, chloroquine, and artemisinin and their byproducts. For example, the widespread resistance to a single therapy (*Alven & Aderibigbe, 2020*), (2) the unfavorable physiochemical side effects, (3) the complex pathogenesis of the (*Okagu et al., 2022*), and the intensive sequestration of the parasites that hinder parasite clearance and reduce the efficacy of artemisinin. Accordingly, recent therapeutic means have been explored to restore the biochemical and physiological norms by killing *Plasmodium* while evading toxic side effects (*Okagu et al., 2022*; *Kluck et al., 2019*). Yet, nano-medicine and medicinal plant extracts appeared to be efficient antimalarial agents (*Patra et al., 2018*; *Laryea & Borquaye, 2019*).

In the current review, we are drawing in the first section a paradigm for the biochemical and physiological changes in blood indicative of malaria in a laboratory panel. In the second section, we scoped the advances in diagnostic techniques to increase the accuracy of detecting malaria, and in the third section, we overviewed the recent therapeutic trends of malaria.

## MATERIALS AND METHODS

The current review manipulated several aspects in malaria disease involving insights in the pathophysiology, the recent diagnostic techniques and points of care, and the updates in the therapeutic trials. The proposed protocol was discussed by the authors. Google Scholar, PubMed, Web of Science, and the Egyptian Knowledge Bank (EKB) were all used to gather articles. To gather articles, a comprehensive screening of all keywords-related studies was done *via* their titles and abstracts. Keywords included malaria diagnosis, malaria

pathophysiology, biochemical changes in malaria, malaria rapid diagnostic tests (RDTs), types of biosensors, molecular diagnosis in malaria, malaria conventional treatment, nanotreatment for malaria, Phyto treatment of malaria, combined treatment of malaria, treatment of malaria relapses. The bulk of referenced scientific papers were in English and conducted online during the period (2015–2023). Yet, when older studies *i.e.,* before 2015 were a certain source of information, we cited them. The inclusion criteria involved peer-reviewed research studies that introduced clinical and experimental applications. Besides, observation and cohort studies were determined. Of note, the comprehensive systematic and narrative reviews that were peer-reviewed, published online, and comprised the aimed keywords were scanned. Case reports were visited by the authors to enhance our understanding. Editorials and studies that were not peer-reviewed, articles with vague abstracts, or unrelated to a certain scientific Database site were excluded.

## RESULTS AND DISCUSSION

### Biochemical and physiological changes

#### Hypoglycemia induced by malarial infection

The disruption of glucose metabolism in malaria chiefly results from the sequestration of intraerythrocytic stages in the arteries of the liver in severe *P. falciparum* infection (*Sengupta et al., 2020*; *Olszewski et al., 2009*; *Tian et al., 2022*; *Glaharn et al., 2018*). Other mechanisms involve the inhibition of ATP-sensitive potassium ($K_{ATP}$) channels, which modulate the pancreatic beta-cell membrane's permeability to potassium. This results in marked calcium influx and insulin release and reduction of the blood glucose level (*Onyesom & Agho, 2011*). In summary, one or more of the following can also contribute to hypoglycemia in malaria:

1. The malaria parasite consumes the host's circulating glucose to meet its energy requirements: The host's glucose supply is a major source of support for the parasites during the asexual phases, which they use to produce ATP through the process of glycolysis (*Sengupta et al., 2020*).

2. The metabolic modifications brought on by parasites that prevent gluconeogenesis: Infection with *P. falciparum* causes the release of cytokines such as tumor necrosis factor (TNF), interleukin (IL)-1, and 6, which block the activity of phosphoenolpyruvate carboxykinase, a crucial enzyme in the gluconeogenetic pathway (*Mavondo et al., 2019*).

3. Malaria-related increases in IL-1 and IL-6 release encourage islet cell hyperplasia resulting in an unregulated insulin response and subsequent reduction of the blood glucose level (*Mavondo et al., 2019*).

4. Erythrocytes can more easily absorb glucose when it is transported across the plasma membrane by GLUT1. The merozoite invasion results in insertion of numerous transmembrane proteins that lead to dramatic change in the plasma membranes of IEs (*Autino et al., 2012*).

5. Additional factors like malaria-related loss of appetite contribute to the downregulation of 5′ AMP-activated protein kinase (AMPK) gene expression, and reduced energy intake (*Apoorv, Karthik & Babu, 2018*). Moreover, the parasite modulated the enzyme 5′

AMP-activated protein kinase (AMPK) and lowered AMPK phosphorylation following infection thus influencing the energetic metabolism of the cell.

### Dyslipidemia induced by malarial infection

Dyslipidemia in patients with malaria is implicated chiefly by the parasite. *Plasmodium* spp. depend on the host for some of the lipids necessary for their growth and development because some of their important lipids cannot be produced *via* intraerythrocytic biosynthesis (*Okagu et al., 2022*). By enhancing lipid absorption, *de novo* synthesis, lowering cellular uptake, and/or other strategies, the parasite manipulates the host's circulating lipid levels to maximize the availability of these vital lipids. The results of a study showed raised TAG levels in the serum, white adipose, and liver, as well as greater free fatty acids and cholesterol in the hepatic tissues of chabaudi rodent malaria (*Autino et al., 2012*). Also, *Kluck et al. (2019)* deduced that *P. chabaudi* infection changes the hepatic mRNA and expression of key enzymes and transcription factors determent in the metabolism of lipids, thus resulting in a lipogenic state. However, variances in the modification of lipid levels in malaria appear to be affected by the severity of the illness and the species of *Plasmodium* parasite. For instance, *P. vivax*-infected patients were observed to have lower TC, LDL, and HDL levels while having higher TAG levels when compared to normal/apparently healthy individuals (*Mesquita et al., 2016*). *Falciparum* malaria has also been associated with decreased levels of HDL, LDL, and total cholesterol as well as increased or unaltered levels of TAGs and very low-density lipoproteins (VLDL) (*Kullu et al., 2018*). Of note, following antimalarial medication therapy, these changes in lipid levels normalized along with a significant decrease in parasitemia and the eradication of clinical symptoms of malaria. Mice infected with *P. berghei* NK-65 in experimental rodent malaria had higher serum lipid profile levels than control mice (*Enechi, Okagu & Ezumezu, 2021*).

*Kidney functions.* *Falciparum* and *Plasmodium malariae* infections are associated with renal manifestations that can produce an immune complex-mediated glomerular disease-causing nephrotic syndrome (*Naqvi, 2021*). *P. falciparum* infection is associated with more renal tubular changes than glomerular changes, and acute renal failure (ARF) is a possible complication (*Lendongo Wombo et al., 2023*). Oliguria, high serum creatinine, and blood urea nitrogen are useful to diagnose ARF. The adverse effects of the malaria parasite on the kidney can cause hypernatremia, hyperkalemia, high blood urea, low urine specific gravity, and metabolic acidosis (*Enechi et al., 2021*).

Severe malaria infection is associated with increased urea and creatinine compared to mild infection. The renal ischemia caused by the sequestration of the parasite into the microvasculature bed of the kidney could be the cause of high urea and creatinine in severe malaria. It was observed that serum protein decreased markedly in severe malaria infection, which is thought to be due to the influence of the parasite on protein synthesis (*Ozojiofor et al., 2020*).

Malaria patients showed a decrease in electrolyte levels. The decrease in $Na^+$, $Cl^-$, and $HCO_3$ levels reflects the level of renal dysfunction, which is proportional to the severity of malaria infection. Low $Na^+$ levels are explained by their loss in urine and sweat to

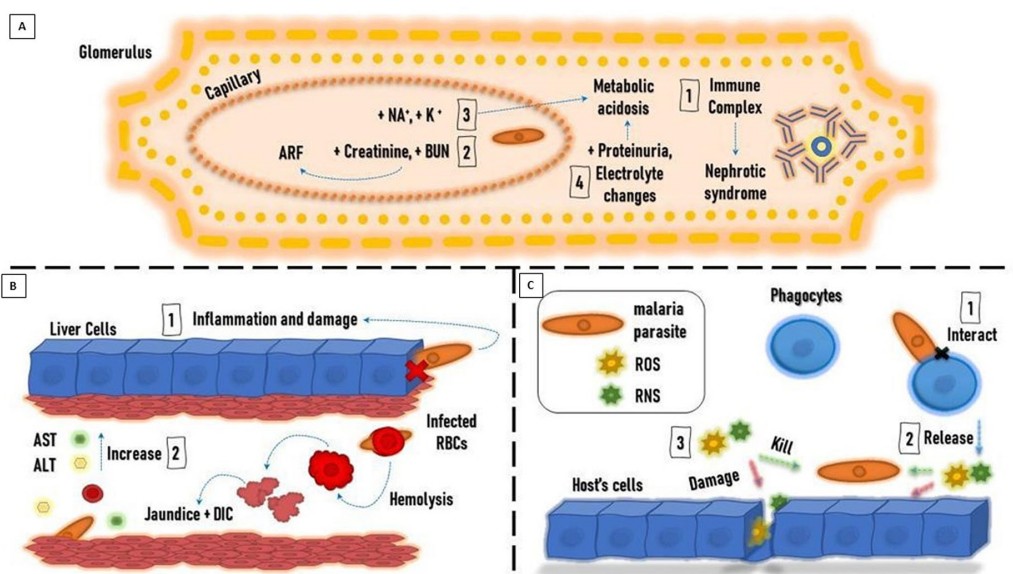

**Figure 1** **Paradigm showing pathogenesis of malaria.** (A) Liver. (B) Kidney. (C) Oxidative stress.

compensate for the high lactase and urea levels of *plasmodium falciparum*-infected patients (*Uwah et al., 2021*). Meanwhile, $K^+$ levels are high in severe infections, with a possible risk of developing metabolic acidosis. Malaria infection causes impaired glomerular filtration, which results in a decrease in Na+ available in the renal tubule for $K^+$ exchange. The resulting imbalance in hydrogen ions causes K+ retention, which can explain the high $K^+$ serum levels (*Enechi et al., 2021*) (Fig. 1A).

*Liver dysfunction.* Malaria infection is usually accompanied by liver dysfunction, which is indicated by the sharp rise in liver enzymes. This can be explained by the invasion of hepatic cells by the parasite and the leakage of liver enzymes into the circulation. The increase in liver enzyme levels is also proportional to the severity of the malaria infection (*Megabiaw et al., 2022*; *Ahad et al., 2022*). Liver dysfunction in malaria cases results from congestion, cellular inflammation, and sinusoidal blockage caused by the parasite (*Auta, Zakariyya & Everest, 2018*).

The incidence of jaundice with severe *P. falciparum* infection ranges from 5.3 to 45% (*Nathumal et al., 2020*). It is caused by hepatic dysfunction, intravascular hemolysis, and microangiopathic hemolysis. This condition is accompanied by high plasma renin activity, high vascular sensitivity to catecholamines, uricosuria, natriuresis, and left ventricular dysfunction (*Ishioka et al., 2020*). Consequently, patients manifesting jaundice usually show complications such as ARF (*Tijjani & Adebayo, 2023*).

Malaria patients showed high conjugated and unconjugated bilirubin levels. On the other hand, serum albumin, globulin, and total protein levels were unchanged. It was reported that high serum bilirubin was the first indicator of liver impairment three days after infection, followed by increased serum aspartate transaminase (AST) and alanine transaminase (ALT) activities (*Bhattacharjee et al., 2021*) (Fig. 1B). Experimentally,

hepatocytes were found to overstimulate hemeoxygenase-1 to degrade free heme to guard against the increased oxidative stress; besides, TNF and NF-$\kappa$B are upgraded thus facilitating neutrophil infiltration and liver damage during the hemolytic phases of acute malaria (*Dey et al., 2019*).

*Oxidative stress.* Reactive oxidant species (ROS) and reactive nitrogen species (RNS) are produced by the immune system of the host in response to the parasitic invasion to kill the parasite. In many cases, the host's cells are damaged by the resulting oxidative stress (OS). It is suggested that OS is a defense mechanism generated by the host against the invading parasite. On the other hand, others argue that OS results from parasite metabolic processes causing complications and organ dysfunction in the host. Of note, degradation of hemoglobin by *Plasmodium* parasites residing inside infected erythrocytes produces free heme, which is released into the bloodstream at the terminal stage of the parasite replication cycle, Further, the oxidative stress intensities on the host might result in complicated life-threatening malaria (*Vasquez, Zuniga & Rodriguez, 2021*).

Total antioxidant capacity is reduced in malaria-infected patients. The activity of catalase (CAT), glutathione peroxidase (GPx), glutathione S-transferase (GST), and superoxide dismutase (SOD) is significantly decreased. There is a decrease in vitamin A, C, and E levels, which are known to be antioxidant vitamins (*Raza et al., 2015*).

Phagocytes are activated as a defense mechanism in response to malaria infections. This causes the production of large amounts of ROS and RNS, which leads to an imbalance between antioxidants and free radicals' formation, triggering OS. It is believed that OS is caused by the increased production of free radicals rather than decreased antioxidant levels. OS represents an important defense mechanism against the parasite (*Gomes et al., 2022*).

ROS produced by phagocytes, including $O_2 \bullet -$ and $ONOO-$, participate in the oxidative destruction of the parasite and the infected erythrocytes. Moreover, proteolytic enzymes and ROS produced by neutrophils can cause apoptosis of endothelial cells. There are high levels of cytokines and endothelial injury associated with severe malaria infection, caused by *P. falciparum*, that can cause organic failure. Therefore, it's believed that the inflammatory processes mediated by the immune response may not be effective against some *Plasmodium* species and can cause harm to the host cells (*Vasquez, Zuniga & Rodriguez, 2021*) (Fig. 1C).

However, it is very important to note that the malaria parasite resides in a highly oxygenated environment (the host erythrocytes) where it produces a huge amount of toxic-free heme imposing oxidative stress on the parasite itself. Yet, malaria parasites tend to tightly manage oxidative stress through active redox and antioxidant defense systems. Heme detoxification protein in *P.falciparum* (*Pf* HDP) is one of the numerous defense mechanisms exerted by the parasite to convert free heme to hemozoin. Interestingly, heme detoxification protein is a protein that exists in the food vacuole of the parasite, the parasitophorous vacuole, and the cytosol of the infected erythrocyte that also acts in the uptake of hemoglobin (*Gupta et al., 2022*).

### Limitations of biochemical and physiological indices

To our understanding, the range of functional reference of any parameter is a resultant interaction of multiple physiological components. In malaria infection, another pathological component becomes added to the stream of biochemical reactions. This might highlight the important role exerted by laboratory professionals to use literature, interrelate data, and facilitate comprehension and prediction of diagnosis *e.g.*, in patients with hypoglycemia and dyslipidemia and who have a travel history to malaria-endemic areas. Another issue, in case clinicians are relying on the physiological and biochemical indices to predict a diagnosis of malaria, it is important to assess sensitivity, specificity, and the rate of device failure to reduce rates of misdiagnosis. More studies for the comparative validity of physiological and biochemical indices for differential diagnosis of malaria infection from other metabolic conditions are still needed.

## Malaria diagnostics techniques
### Rapid diagnostic tests or immunochromatographic tests

This type of device is considered a widespread point-of-care test (POCT) to diagnose malaria. Rapid diagnostic tests (RDTs) is a lateral flow immunoassay technique that detects the presence of biomarkers specific to the *Plasmodium* parasite. The technique is a single-step test with the advantage of being cheap and fast. However, the affordability and accessibility RDTs do not define them as distinctive malaria diagnostic factors (*Wittenauer, Nowak & Luter, 2022*). This device is comprised of a nitrocellulose membrane netted with antibodies and antibodies against the target antigen to detect several biomarkers.

Commercially, the available RDTs biomarkers involve *P. falciparum* histidine175 rich protein-II (Pf-HRP-II), which is water soluble and is expressed on the surface of the RBCs. Also, plasmodial aldolase (pALD), as well as plasmodial lactate dehydrogenase (pLDH), are parasite-specific glycolytic enzymes recognized in all plasmodium species. Yet, several drawbacks have been determined for RDTs. For instance, Pf-HRP-II is the most principal target to detect the parasite; yet, it is exclusively produced by *P. falciparum* and thus not sensitive for other species, has a detection limit >40–100 parasites per μL, persists as positive for up to 31 days post-treatment, reported for cross-reactivity either with other plasmodium species or autoantibodies; besides it cannot detect viability of the parasite. Regarding HRPII, it is released during schizogony into the bloodstream and remains positive for several weeks, even after the clearance of infection, shows a higher detection limit of parasites/μL, and is inefficient in differentiating parasite species (*Grandesso et al., 2016*). The pALD exhibits comparable drawbacks similar to HRPII, yet, its positive result defines the viability of the parasite.

The limitation of conventionally used RDT also involves the reduction in peripheral parasitemia owing to *Plasmodium* sequestration. Also, PfHRP2 antigen are also useful in Africa, India, Asia, middle east, Amazon regions in addition to sub-Saharan countries and South America, since some isolates lacking this antigen have been found from these areas. The the presence of Pf-HRP-II/III deletions in the parasites reinforces the necessity for novel diagnostic techniques for systematic surveillance in endemic regions (*Berhane et al., 2018*).

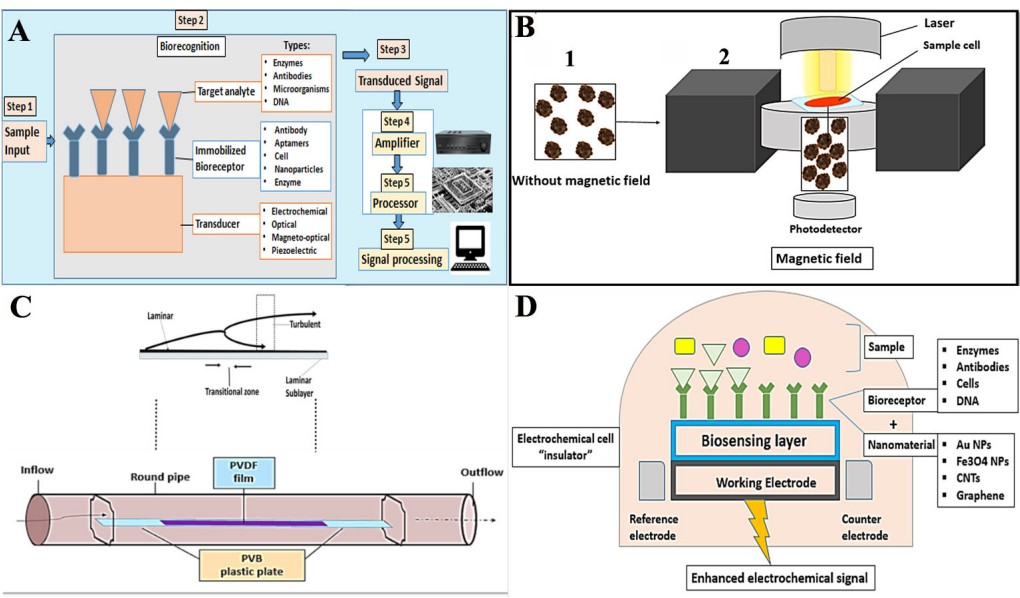

**Figure 2   A general paradigm for biosensors.**

### Biosensors as a recent "point-of-care"

Firstly, what are the biosensors? Biosensors are detector devices composed of immobilized bio-receptors (bio-recognition element) and a transducer. The bio-receptor might be an antibody, enzyme, DNA, or microorganism that detects and bind with the target analyte, and the produced changes in the physicochemical characteristics (*e.g.*, electrochemical, magnetic, optical, and piezoelectric acaustic) are transformed into a quantitative or semi-quantitative electrical signal that can be measured by a transducer as shown in Fig. 2 (*Ding, Srinivasan & Tung, 2015*). Such devices were designated to explore for analytes like hemozoin and several other enzymatic markers *e.g.*, LDH, glutamate dehydrogenase (GDH), merozoite surface protein 3 (MSP-3), and ALD in enzyme-based biosensors (*Dutta, 2020*).

### Types of biosensors

*Magneto-optical biosensors.*  The hemozoin is a paramagnetic structure that was found to possess a small and positive magnetic moment to externally applied magnetic fields. Based on this unique property, *Mens et al. (2010)* determined that when a malaria-infected blood sample was placed in a magnetic field hemozoin crystals aligned within the field. Yet, crystals became free when the field was removed as shown in (Fig. 2B).

*Piezoelectric acoustic transducer.*  The poly-vinylidene fluoride (PVDF) film is used as a smooth thin plate that provides no-slip condition. When a viscous blood sample propagates along this smooth thin plate, a thin shear layer is formed near the surface. This layer can be divided into three zones according to the different flow states: (1) the laminar flow zone, (2) the transition zone, and (3) the turbulent zone. Fluid motion produces hydrodynamic

noise due to the vibration of the PVDF film. The intensity of the flow noise increases as the speed of the fluid flow increases (*Li et al., 2019*). Therefore, the developed acoustic transducer can be capable of assessing the velocity of the fluids within capillary vessels. Yet, future work is necessary to improve the characteristics of this sensor to be a reliable diagnostic tool for malaria in blood specimens (*Katta & Sandanalakshmi, 2021*) (Fig. 2C).

*Electrochemical biosensors.* This type of biosensor perceives an electrical signal when a target biomarker (analyte) binds with the specific bioreceptor. The concentration of the target analyte is directly related to the intensity of the electrical signal, Fig. 2D. It shows high sensitivity, reasonable cost, requires small specimens, quantitative and feasible. Nevertheless, these are thermos-sensitive, with a narrow temperature range. The limited shelf time and nonspecific reactions are also other challenges for these biosensors (*Menon et al., 2020*).

In an attempt to identify $\beta$-hematin (synthetic hemozoin), *Obisesan et al. (2019)* processed an electrochemical nano-sensor that was composed of metal oxide of copper, aluminum, and iron (as nanoparticles) deposited on the gold electrode. So far, the maintenance of a stable electric current and electrodes after several cycles, and the high-cost technology might hamper the method.

Graphene is another example that detects the reactions of the electronic transfer of hemoglobin from the ferrous state ($Fe^{++}$) into the ferric state ($Fe^{+++}$) to produce hemozoin (*Moutaouakil, Belmoubarik & Peng, 2020*; *Hole & Pulijala, 2021*).

*Optical biosensors.* This type of biosensor is established on identifying alterations in light of the reaction between the target analyte and the bio-recognition (bio-receptor) element. The most popular types are surface plasmon resonance (SPR) and fluorescence biosensors.

*SPR biosensors.* These sensors detect the reaction between the bio-receptors and the target analyte through the alteration in the refractive index of the plasma resonance substrate, the SPR angle, and the intensity of light reflectance. These types of biosensors are label-free, have high sensitivity and resolution, permit real-time evaluation, and possess a high signal-to-noise ratio; hence can be considered suitable for point-of-care applications of malaria disease. However, SPR bio-sensors are motion-sensitive and rely on the quality of the light indicator. They entail the adjustment of light distance and angle and are affected by the molecular size and concentration of the analyte. Therefore, full automation is necessary to avoid extensive calibration and light intrusions (*Ragavan et al., 2018*) as in (Fig. 3A).

There are several models for this type; for example, *Chaudhary, Kumar & Kumar (2021)* attempted to improve the SPR using two sheets of air holes in the form of a hexagonal lattice (photonic crystal fiber) and a tinny layer of gold (Au). Evaluation of this model showed a recognizable shift in the SPR resonance wavelength ($\lambda$) and refractive index with malaria-infected red blood cells (RBCs) compared with normal RBCs. In addition, it was useful to discriminate early infections and the different stages of the parasite; for instance, ring stage at $\lambda = 13{,}714.29$ nm/RIU, trophozoite at $\lambda = 9{,}789.47$ nm/RIU, and schizont at $\lambda = 8{,}068.97$ nm/RIU (*Chaudhary, Kumar & Kumar, 2021*).

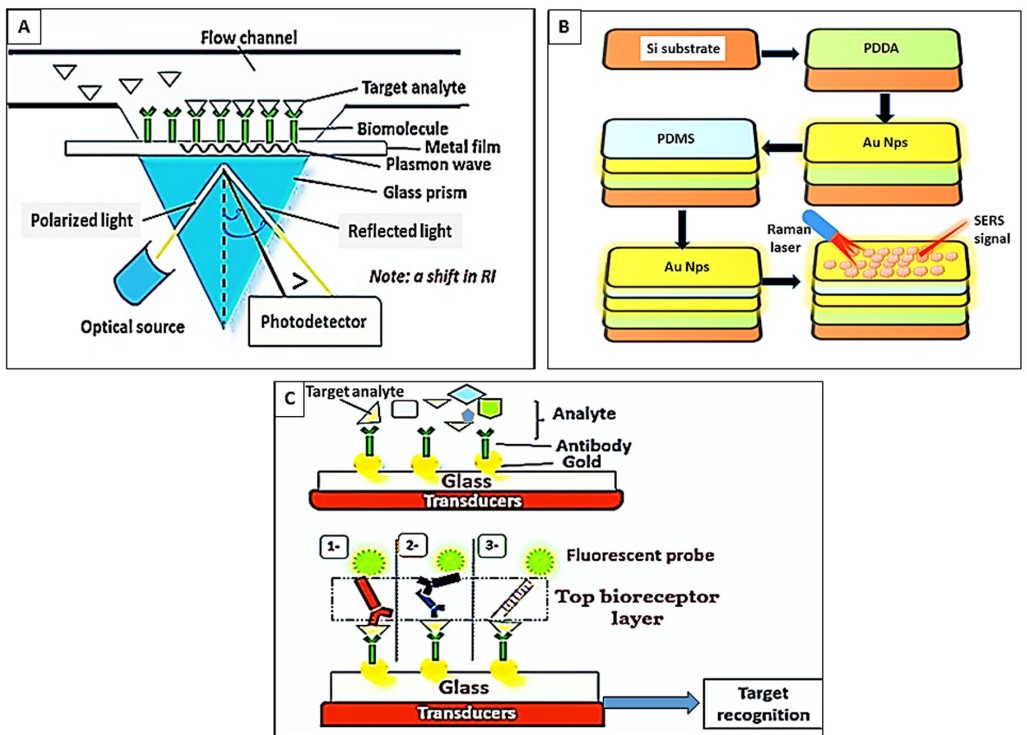

**Figure 3** **Paradigm of optical biosensors.** (A) SPR biosensors. Target analyte reacted with immobilized bioreceptors. Excitation of electrons in the metal film by inclined photons. Dissemination of electrons parallel to the metal surface (plasmon wave). Shifting in the refractive index (RI). (B) Surface-enhanced Raman spectroscopy. (1) Silicon (Si) substrate, (2) coating of Si substrates by polydi-allyl-dimethyl ammonium (PDDA), (3) assemblage of AuNps on PDDA, (4) formation of the poly-di-methylsiloxane (PDMS) layer, (5) embedding of Au Nps on PDMS, (6) hematin detection by Raman laser. (C) Fluorescence biosensors. The target analyte binds the bottom immobilized bioreceptor layer. Then a "Top bioreceptor layer" is detected by either (1) Abs- fluorescently-labeled Abs, (2) a double layer of antibodies (Abs)- fluorescently-labeled Abs, (3) or a monolayer of fluorescently-labeled aptamers.

*Surface-enhanced Raman spectroscopy (SERS) and enhanced Raman signals.* Raman spectroscope relies on the very few photons that *elastically* scatter and interact with the chemical bonds of the molecules or the functional groups within a substance. In malaria, hemozoin exhibits Raman scattering. It is a popular highly sensitive and specific optical technique that however possesses a low signal-to-noise ratio (*Villena Gonzales, Mobashsher & Abbosh, 2019*; *Xu et al., 2022*). Several methods were developed to increase the efficacy of the SERS biosensors, for example, (1) the manipulation of the Graphium weiskei butterfly rings that interact with hemozoin (*Garrett et al., 2014*), (2) the usage of a magnetic field that brings the nanoparticles and the paramagnetic $\beta$-hematin into line for the laser spotlight to intense the Raman signal (*Yuen, 2012*), (3) manipulation of gold nanoparticles (Au Nps) for their plasmon coupling features and near-infrared SPR to improve the Raman signals and detection of haematin (*Cai et al., 2021*) as shown in (Fig. 3B).

*Fluorescence biosensors.* These sensors rely on the emission of fluorescent light from fluorophore molecules at a definite wavelength (λ) when the bio receptor interacts with the target analyte the intensity of light is correlated to the concentration of the target analyte (*Ragavan et al., 2018*). For example, *Minopoli et al. (2021)* evaluated a fluorescence biosensor composed of immobilized Au Nps onto a glass substrate. The target analyte binds to the bio-receptor layer (antibodies), which is covalently attached to the surface of Au Nps. Thereafter, a top bioreceptor layer of fluorescently labeled aptamers is applied in a sandwich pattern to recognize the reaction as shown in (Fig. 3C). The suggested biosensor specifically detected *P. falciparum* lactate dehydrogenase enzyme in the whole blood down to 0.3 ng/mL without any sample pre-processing (*Minopoli et al., 2021*).

### The lab-on-a-chip and micro-device technology

The novel microdevices, especially lab-on-a-chip devices, have been considered a potential portable point of care for the diagnosis of malaria. This technology relies on the concentration of RBCs through their margination using micro-channels for more specific and sensitive detection. The design of the micro-channels simulates blood capillaries with a caliber of less than 300 μm. It is well known that RBCs are more deformable and smaller than white blood cells (WBCs). Therefore, RBCs under normal conditions flow faster than WBCs along the axial center of the blood capillaries; whereas, WBCs marginate to the endothelium of the blood capillaries. In malaria infection, the stiffer infected RBCs act similarly to WBCs and show cytoadherence to the endothelium (*Giacometti et al., 2021*).

In the fabricated micro-channel, *P. falciparum*-infected RBCs align along the sidewalls "*via margination*"; thereafter, infected RBCs can be removed and separated using a three-outlet system as shown in Fig. 4A (*Kolluri, Klapperich & Cabodi, 2018*). Another model was invented that relied on the magnetic properties of the hemozoin to fasten the margination of the infected RBCs in a magnet-based microfluidic device. The device is coupled to a nickel wire that attracts infected RBCs and a permanent magnet to create an external field (∼0.6 T). Thus, allowing the separation of malaria-infected RBCs as in Fig. 4B (*Nam et al., 2019*).

## Multiplex polymerase chain reaction

Molecular detection of Plasmodium DNA in blood is a sensitive diagnostic tool (5 parasites/μL) (*Pham et al., 2018*). Application of multiplex polymerase chain reaction (PCR) determined the significant existence of asymptomatic reservoirs that may prolong the endemicity of malaria. For instance, a study in Eastern parts of Afghanistan reported the presence of *P. falciparum*, *P. vivax* and their co-existence in asymptomatic individuals (*Mosawi et al., 2020*). Another study in Malaysia demonstrated that in 251 samples mono-infection of *P. vivax* was reported in 39 cases (16%), *Plasmodium falciparum* in 50 cases (20%), *P. vivax* in 39 cases (16%), *P. knowlesi* in 9 cases (4%), and mixed infections in 20 cases (8%) (*Jiram et al., 2019*).

*Multiplex human malaria array.* A new sensitive generation of rapid diagnostics for the quantification of malaria antigens may facilitate the screening of asymptomatic malaria. The Q-Plex™ Human Malaria Array (Quansys Biosciences, Logan, UT, USA) has been evaluated

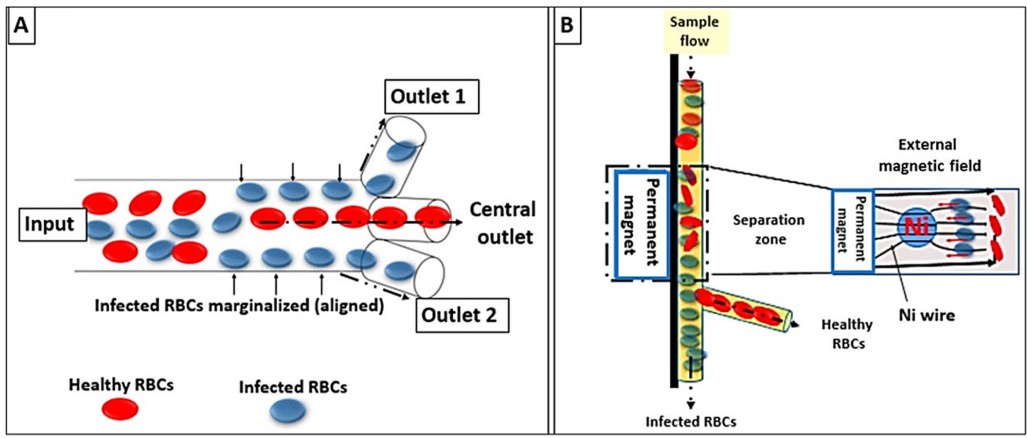

**Figure 4** **Lab-on-a-chip and microfluidic system designs.** (A) Infected RBCs are displaced to the margins of the micro-channel wall (RBCs margination) whereas healthy RBCs migrate to the axial center due to gradients in the flow velocity. (B) Faster margination of infected RBCs was achieved by an externally applied magnetic field.

to quantify the antigens specific for *Plasmodium* spp. *e.g.*, HRP-2, *Pf*-LDH, *Pv*-LDH, and Pan-LDH. The new kit showed 99.5 improved specificity whereas sensitivities against PCR were 92.7%, 71.5%, 46.1%, and 83.8% for HRP2, *Pf*-LDH, *Pv*-LDH, and Pan LDH, respectively (*Jang et al., 2020*).

## Loop-mediated isothermal amplification PCR

Loop-mediated isothermal amplification assay (LAMP) is a recently-developed simple, field applicable, molecular malaria diagnostic technique that employs amplification of nucleic acids under isothermal conditions. Given the advantages of minimum equipment, short time, 99% sensitivity and accurate diagnosis of asymptomatic malaria, LAMP is now considered a superior screening test than conventional methods for malaria diagnosis especially in non-endemic countries (*Feleke, Alemu & Yemanebirhane, 2021*). Considering 18s rRNA as the target gene, SYBR green I also adds higher specificity to LAMP than the nested PCR (*Lai, Ooi & Lau, 2021*).

## Rolling circle enhanced enzyme activity detection

Rolling circle enhanced enzyme activity detection (REEAD) is the process of detecting endogenous enzyme activity at a single-catalytic event. REEAD was shown to assess the enzyme activity in both *Plasmodium* parasites and single human cells. The rolling circle amplification products (RCPs) are detected using "an organic dye-tagged linear DNA probe". Nevertheless, the high cost hampers its usage on a wide scale. A novel class of activatable probes, NanoCluster Beacons (NCBs), has been shown to facilitate REEAD assays by being cost-effective, sensitive, photostable, easily prepared without purification and capable of producing a large fluorescence enhancement upon hybridization (*Juul et al., 2015*). *Givskov et al. (2016)* deduced the assessment of topoisomerase -I from *P. falciparum* parasite is a useful tool for detecting malaria in epidemic areas. Combining

rolling circle amplification techniques with affinity biosensors using optical probes is a promising recently-tailored research area (*Schmidt et al., 2022*).

## Micromagnetic resonance reflaxometric tests

Micromagnetic resonance reflaxometric (MMR) tests is a novel procedure for sensitive, quantitative and rapid detection of *Plasmodium spp.*-infected RBCs with minimal sample preparatory steps and without any chemical or immunolabeling.

A parasitemia level of fewer than ten parasites per microliter in a volume below 10 µl of whole blood is detected in a few minutes by the means of magnetic field.

In the course of the intraerythrocytic cycle of malaria, the parasite metabolizes large amounts of cellular hemoglobin and converts it into hemozoin crystallites. On exposing intact blood cells to a powerful magnetic field, the paramagnetic $Fe^{3+}$ ions in hemozoin distract the synchronizing hydrogen atoms. The more the hemozoin crystals inside the RBC's, the more rapidly the hydrogen atoms are disrupted reflecting the 'parasite load' in blood (*Kumar & Renu, 2015*).

## Recent malaria therapeutic perspective

A good antimalarial agent not only eliminates the *Plasmodium* parasites but also rectifies pathophysiology and altered biochemical parameters associated with the disease (*Enechi et al., 2021*). Control of malaria is chiefly dependable on chemotherapy involving quinine, amodiaquine, 4-aminoquinolines, primaquine, pyrimethamine, sulfonamides, mefloquine, lumefantrine, halofantrine, atovaquone, proguanil, artesunate, artemeter, piperaquine, and chloroquine. Chloroquine is a competent antimalarial drug. Yet, artemisinin and its semi-synthetic derivatives are more recommendable owing to their effectiveness compared with quinine. Therefore, they are the current first-line treatment against *P. falciparum* malaria. Nevertheless, limitations for non-artemisinin drugs involve low efficacy, high cost, toxicity, short half-life time, and low water solubility. In addition, to reach the desired drug concentration exposure to undesired side effects and non-specific drug targeting frequently occur. In addition, the current antimalarial drugs are facing the emergence of resistance which prevails owing to their similar chemical structures. For instance, the cross-resistance that occurs between 4-aminoquinolines, CQ, and AQ. Currently, artemisinin-resistant *Plasmodium* parasites are conquering some regions of the world, therefore, research for more efficient and safer alternatives is demanded (*Erhirhie et al., 2021*). Table 1 shows comparative insights into conventional, phytotherapy, and nano-medicine.

### Chemical falcipain inhibitors

Falcipains are cysteine protease enzymes related to the papain family. The Papain family of proteins has a wide range of aminopeptidases dipeptidyl peptidases, endopeptidases, and enzymes with both exo- and endopeptidase action. Falcipain-2 and -3 are key protease enzymes produced by the *P. falciparum* parasites and are involved in the hydrolysis of hemoglobin and the production of free amino acids. Falcipain-2 is convoluted in slicing band 4.1 protein and ankirin involved in the cytoskeleton of the red blood cell. Interestingly, other *Plasmodium* species express proteases homologs to falcipain-1, -2 and-3 proteases. Therefore, falcipain inhibitors are regarded as promising agents against other
**Table 1  Conventional treatment, phytotherapy, and nanomedicine.**

| | Conventional treatment | Phytotherapy | Nanomedicine | References |
|---|---|---|---|---|
| Source/origin | **4 major classes:**<br>• Quinoline-related compounds (plant origin)<br>• Antifolates,<br>• Artemisinin derivatives (plant origin),<br>• Two antibiotic families: macrolides and tetracyclines<br>• **Recently,** Falcipain inhibitors *e.g.,* Quinoline-4-carboxamide derivative, The (E)-chalcone inhibitor, and Tetracycline | Stem, seeds, leaves, or the roots of the medicinal plant<br>**Examples:** *Buchholzia coriacea,* Gymnema inodorum, *Anogeissus leiocarpus, Triumfetta cordifolia, Bidens pilosa, Syzygium guineense* and *Parinari congensis,* Amaranthaceae, Annonaceae, Nyctagynaceae, Rubiaceae, Vitaceae *etc.* | • **Metallic NPs**<br>*e.g.,* gold and silver<br>a. Non-biological methods (physical and chemical).<br>b. Biological method (green NPs from bacteria, fungus, plant)<br>• **Inorganic nonmetallic NPs**<br>a- Titanium dioxide,<br>b- Zinc oxide,<br>c- Cadmium oxide<br>• **Carbon-based NPs**<br>a- Multi-walled carbon nanotubes.<br>b- Carbon–silicon | *Erhirhie et al. (2021)*;<br>*Boonyapranai et al. (2022b)*;<br>*Dahiru, Badgal & Neksumi (2023)*;<br>*Ezenyi et al. (2020)*;<br>*Panneerselvam et al. (2019)* |
| Action | • **Quinoline**: interacts with RBCs membrane stomatin protein<br>• **Antifolates**: antagonize vit B9 (folic acid)<br>• **Artemisinin**: decomposes the endoperoxide bridges of heme producing toxic anti-parasite free radicals.<br>• **Macrolides**: inhibits parasite's RBC invasion<br>• **Tetracycline**: targets Plasmodium apicoplast<br>• **Falcipain inhibitors**: hydrolyses hemoglobin | • **Buchholzia coriacea** Engl. seeds and **Gymnema inodorum leaf**. improves the hematological and biochemical parameters.<br>• *Anogeissus leiocarpus.* Anti-oxidants improve liver functions.<br>• **Triumfetta cordifolia,** *Bidens pilosa, Syzygium guineense* and *Parinari congensis extracts*. anti-plasmodial activity. | • Diffusion *via* RBCs membrane,<br>• or interacts with the RBC membrane | *Gaillard et al. (2016)*;<br>*Wilson et al. (2015)*;<br>*Biddau & Sheiner (2019)* |
| Advantage | • Artesunate of choice in adult cerebral falciparum malaria<br>• The artemether is rapidly absorbed<br>• Artemisinin selectively toxic to parasite | • Ameliorates physiological and biochemical changes<br>• Selective toxicity of the parasite | • Biological extraction of metal NPS is friendly to the ecosystem.<br>• Inorganic NPs are non-toxic, hydrophilic, biocompatible, and stable compared to organic materials<br>• Improved AUC curve.<br>• Lower doses<br>• Lower frequencies of administration,<br>• Improved half-life time,<br>• Better water solubility | • *South East Asian Quinine Artesunate Malaria Trial (SEQUAMAT) group (2005)*;<br>• *White, van Vugt & Ezzet (1999)* |

*(continued on next page)*
**Table 1** (*continued*)

| | Conventional treatment | Phytotherapy | Nanomedicine | References |
|---|---|---|---|---|
| **Limitations** | • Multi-drug resistance,<br>• Declining efficacy,<br>• High cost,<br>• Toxicity,<br>• Short half-life time,<br>• Low water solubility<br>• Limited as chemopro-phylactic e.g., Quinine<br>• Restricted in organ dysfunction e.g., hepatic dysfunction →impairs the conversion of quinine to 3-hydroxyquinoline. | • Parasiticidal efficacy needs further research.<br>• Systemic effects of the medicinal plant not evaluated.<br>• Effects are dose-dependent. | • Non-biological extraction of metallic NPs is hazardous.<br>• Limited industrial chances,<br>• Not cost effective,<br>• Limited clinical trials | • *Baruah et al. (2018)*<br>• *Shanks (2016)*<br>• *Erhirhie et al. (2021)* |
| **Prospective Areas of Research** | • Studies on gene mutation,<br>• Trials of combined treatment with modified chemical formulas | • Effects of combined crude plant extracts + chemotherapy against cytokine storm of acute malaria<br>• Phyto therapies versus chemotherapies regarding anti-inflammatory, and anti-oxidative effects using different plant species.<br>• Evaluation of the *plasmodocidal* effects of various plant species in particular in complicated and resistant strains.<br>• Refining research on a species-specific level.<br>• Assessing its antimalaria prophylactic activity. | • The biological synthesis of metallic NPs (green NPS)<br>• Comparative studies of Green NPs using different plant species<br>• Assessing solid lipid NPs as a carrier or antimalaria target therapy in resistant malaria<br>• Systemic assessment of the biological effects of NPs<br>• Evaluation of NPs with or without drugs in complicated or acute malaria. | |
parasite species (*Rosenthal, 2020*; *Ettari et al., 2021*). Examples involve (1) quinoline-4-carboxamide derivative. This compound was developed using molecular hybridization. The efficacy of quinoline-4-carboxamides involves the inhibition of falcipain-2; besides altering the food-vacuole and morphology of the parasite (*Singh et al., 2021*). (2) The (E)-chalcone inhibitor. Chalcones are organic compounds with antimalarial and antioxidant activities. E-chalcone 48 was found to bind only to the falcipain-2-substrate binding cleft; thus, achieving high specificity (*Machin, Kantsadi & Vakonakis, 2019*). (3) Tetracycline. Tetracycline is a broad-spectrum antibiotic; where its derivatives were found to inhibit Falcipain-2 by interacting with its distal allosteric site (*Hernández González et al., 2021*).

### Plant extracts

Medicinal plants proved to have impressive anti-plasmodial activities. Interestingly, studies afford the scientific basis of plant extracts and their adventitious usage in folkloric herbal medicine. Herein, we presented some examples (*Laryea & Borquaye, 2019*). Examples involve (1) **Buchholzia coriacea Engl. Seeds**. In a murine study, a flavonoid-rich extract of *B. coriacea* seeds (FEBCS) had a dose-dependent chemo-suppression; yet its efficacy was lower than combisunate. FEBCS improved the hematological and biochemical parameters to near-normal levels with no morphological or behavioral sign toxicity (*Enechi et al., 2021*). (2) **Gymnema inodorum Leaf**. In murine models, this plant extract exhibited protective effects against the biochemical changes due to *Plasmodium berghei e.g.*, hypoglycemia, dyslipidemia, and acute liver and Kidney injury (*Boonyapranai et al., 2021a*); in addition, alterations in hematological (*Ounjaijean et al., 2021*) and blood coagulation parameters were normalized (*Boonyapranai et al., 2022b*). (3) **Ethanol extract of the stem bark of *Anogeissus leiocarpus* (EESAL)**. Treatment of malaria-infected murine models with EESAL showed improved hematological parameters, restored biochemical indicators for lipid peroxidation and liver functions, and exerted antioxidant effects. Besides, EESAL was tolerable per oral up to 5,000 mg/kg body weight (*Dahiru, Badgal & Neksumi, 2023*). (4) **Ethanol extract of Triumfetta cordifolia**. A Nigerian study revealed the efficient anti-plasmodial activity of *T. cordifolia*; yet, its mechanism of action is still not recognized (*Ezenyi et al., 2020*). (5) **Bidens pilosa, Syzygium guineense and Parinari congensis extracts**. In mammalian cell lines, these extracts exhibited very feeble to no cytotoxicity. Yet, it showed high selectivity for *Plasmodium* parasites. Extracts from *Syzygium guineense* and *Parinari congensis* were the most toxic against the parasite (*Laryea & Borquaye, 2019*).

### Nano-medicine

Why is nano-medicine regarded as the new era in malaria treatment? *Plasmodium* parasites evade the host immune system within the RBCs. Thus, targeting the parasite necessitates the ability of the drug to cross membranous barriers of the RBCs and the parasitophorous vacuole and plasma membrane of the *Plasmodium* parasite. Thereafter, the drug, in an optimum dose, should reach a definite target *e.g.*, a food vacuole or an apicoplast membrane of the parasite (*Anamika et al., 2020*). Yet, nanocarriers are less than 1,000 nm with a high surface area: volume ratio; thus, allowing for improving the area under curve (AUC curve) and efficacy of the drugs with lower doses and frequencies of administration, improved half-life time, and water solubility (*Rathee et al., 2015*; *Patra et al., 2018*).

Overall, antimalarial drugs can approach their targets either through the lipid membrane of the RBCs by diffusion or through interacting with the protein component of the erythrocyte membrane (receptors and transporters) by carrier-mediated transportation. Carrier-mediated transportation is either passive or active (*Clemons et al., 2018*). In the case of passive carrier-mediated transportation energy is not required; hence considered facilitated diffusion. This type is attained by the conventional nanocarriers (the liposomes, the polymeric nanoparticles, and the long-circulating polyethylene glycol nanocarriers (PEGylated)). Nanocarriers, which are less than 80 nm in diameter, leak through the porous membrane of the infected RBCs. In this case, the nano-carrier bypasses the cytosol of the RBCs and sinks directly to the parasite through the tubovesicular membrane (TVM). TVM is an alteration in the architecture of the RBCs wherein a network is synthesized by the parasite to connect the membrane of the RBCs with the membrane of the parasitophorous vacuole (that surrounds the parasite inside the RBCs). This TVM facilitates *Plasmodium*-derived proteins transport, nutrition uptake, and elimination of waste products outside the cell; thus, allowing the thriving of the parasite (*Rai et al., 2017*). For example, human serum albumin (HSA) has been recognized as a potential drug-delivery vehicle for artemether (ATM) in models infected with *P. berghei* species. These nanoparticles showed two-fold higher peaks in the drug concentrations inside the RBCs. Overall, HSA is non-toxic and non-immunogenic (*Kaur et al., 2021*). In addition, HSA is an endogenous transport protein with several drug binding sites; thus, acting as a useful drug carrier to improve stability and half-life time of the loaded drugs or perform as a targeting agent. HSA has high specificity and uptake by the parasitized RBCs (*Tahir, Malhotra & Chauhan, 2003*). In 2019, *Esu et al. (2019)* speculated that HSA can intensely bind to ATM and its byproducts by thiol and amino groups, and thus can be used in drug-resistant.

On the contrary, active drug targeting requires energy consumption and surface activation of the nanocarriers by using specific ligands that bind to receptors on the membrane of the infected RBCs. These ligands might be in the form of antibodies, carbohydrates, peptides, or proteins. In this type, the drug is delivered to the parasite through the cytosol of the RBCs (*Anamika et al., 2020*). For instance, pyronaridine and atovaquone can be formulated as immunoliposomes and interact with the glycophorin-A receptor on the RBCs through anti-glycophorin A-antibodies (*Biosca et al., 2019*). Artemisinin can be loaded on nanostructured lipid carriers and react with the transferrin receptor on the brain cells (in cerebral malaria) through transferrin ligands (*Emami, Yousefian & Sadeghi, 2018*). Recently, the emergence of the polymeric wall in the encapsulation of lipophilic (oil) drugs has been introduced. These Nanocapsules have high entrapment efficacies and low toxicity and polymer content. In addition, it increases the solubility of the compounds and evades drug inactivation in the gut (*Rajabi & Mousa, 2016*). In a prior study, the poly (*butyl methacrylate-co-morpholino ethyl sulfobetaine methacrylate*)-based nanoparticles were shown to specifically target *P. falciparum*-infected RBCs compared with the healthy RBCs by a ratio 74.8%: 0.8% (*Biosca et al., 2021*).

*Green silver nanoparticles (Ag NPs).* Green synthesis is the processing of biologically friendly elements *e.g.*, plants for nanoparticle synthesis. These are other recent models with

high therapeutic potential against malaria. For example, silver nanoparticles extracted from the leaves of the Indigofera oblongifolia plant alleviated the oxidative injury of the infected livers and improved ISHAK's inflammatory scale (or the modified histology activity index score) at a dose of 50 mg/kg for 7 days (*Dkhil et al., 2020*). The neem-AgNPs were shown to have no hemolytic activity against healthy and parasitized RBCs (*Ghazali, Mohamed Noor & Mustaffa, 2022*).

*Solid lipid nanoparticles.* These are produced as an alternative to liposomes, emulsions, polymeric nanoparticles, and lipid nanocarriers and is considered the second generation of lipid carriers (*Vanka et al., 2018*). For instance, Chloroquine was structured in the form of solid lipid nanoparticles that react with glycosaminoglycans (GAG)-like receptors on the membrane of the RBCs through specific ligand (heparin). The results showed higher efficacy with less adverse effects compared with the conventional treatment (*Muga, Gathirwa & Tukulula, 2018*).

## Models for anti-malaria nano treatments

Using *in vitro* model, nanosized solid dispersion of chloroquine and primaquine loaded with dissolving microarray patches enhanced treatment *Plasmodium vivax infection* (*Anjani et al., 2023*). In BALB/c mice, the synthesized *nano-* chloroquine revealed its appropriate size and solubility. Highest suppressive impact of nano-chloroquine on the growth of *Plasmodium* parasitic was determined at 16 mg/kg dose eliminating 95% of the parasites. $ED_{50}$ (the dose of drug that results in a precise effect in 50% of the experimental treated group) is detected at a dose of 7.7 mg/kg. Biochemical indices revealed that the synthesized nano-chloroquine was safer than chloroquine. Moreover, no adverse effects were detected when assessed in tissues (*Elmi et al., 2022*). Of note, *Usman & Farrukh (2018)* formulated the polymeric iron nano-chloroquine phosphate using the polyol method and showed that its nanoparticle size is approximately 10 nm. Using rodent models infected with *P. berghei*, nano-chloroquine ameliorated the oxidative damage in the mitochondria of the liver and spleen (*Tripathy et al., 2013*).

Assessing the antimalarial efficacy of primaquine phosphate loaded on polyethylene glycol galactosylated nano-lipid carriers showed suppression of parasitemia by 99.46% at a dose of 2 mg/kg/d. After the 35-day post-treatment, there was better inhibition of the parasitemia by 28% and a higher survival rate (66.66%) when compared with pure drug (*Baruah et al., 2018*). Also, zein nanoparticles as carriers of artemether have been suggested for the treatment of severe malaria as a pioneering intravenous dosage form (*Boateng-Marfo et al., 2021*).

Yet, manipulating other anti-malaria drug classes (Table 1) using different forms of nanoparticles (Tables 1, 2 and 3) to attain therapeutic efficacy particularly during complicated and severe malaria is yet to be investigated. Also, translating antimalaria nano-medicine from laboratory trials to clinical trials would improve intensely our insights into the dosage, frequency, and treatment duration appropriate to treat malaria.

**Table 2  Types of nanocarrier-mediated transportation through erythrocytic membrane.**

| Front of comparison | Passive carrier-mediated transportation | Active carrier-mediated transportation |
|---|---|---|
| Type of NPs | Conventional nanocarriers with a diameter of 80 nm. | Nanocarriers with specific ligands |
| Examples | NPs in the form of:<br>• Liposomes,<br>• polymeric nanoparticles,<br>• or long-circulating polyethylene glycol Nanocarriers (PEGylated) | Ligands in the form of:<br>• Antibodies<br>• carbohydrates,<br>• peptides,<br>• or proteins |
| Application of drug-NPs | Human serum albumin as drug-delivery Artemether in *P. berghei* infection | • Pyronaridine and atovaquone (formulated as immunoliposomes) acts via anti-glycophorin A-antibodies →interact with the Glycophorin-A receptor on the RBCs<br>• Artemisinin (loaded on nanostructured lipid carriers) acts via transferrin ligands → react with the transferrin receptor on the brain cells (in cerebral malaria). |
| Energy requirement | ATP is not required (facilitated diffusion). | ATP is required |
| Mechanism of action | Bypasses the cytosol of the RBCs and sinks directly to the parasite through the TVM. | Nanocarriers via specific ligands + receptors on the membrane of the infected RBCs → membrane surface activation of infected RBCs → The drug is delivered through the cytosol of the RBCs |
| Effects | • Two-fold increases in the drug concentrations inside the RBCs.<br>• Non-toxic and non-immunogenic.<br>• HSA is an endogenous transport protein → several drug binding sites.<br>• Improved stability and 1/2 lifetime of the loaded drugs<br>• Acts as a targeting agent.<br>• High specificity<br>• High uptake by the parasitized RBCs | • High entrapment efficacies<br>• Low toxicity and polymer content.<br>• Increased solubility<br>• Evades drug inactivation in the gut<br>• High target specificity |

**Table 3  Green silver NPs and solid lipid nanoparticles (NPs) in malaria treatment.**

| | Green silver nanoparticles (AgNPs) | Solid lipid NPs |
|---|---|---|
| Origin | Extracted from the plants | 2nd generation of lipid carriers |
| Examples | • The Indigofera oblongifolia plant<br>• Neem-AgNPs | Chloroquine-solid lipid NPs |
| Effects | Antioxidative, anti-inflammatory, anti-hemolytic | React with GAG-like receptors on the RBCs |

## Limitations of nano-mediated drug delivery systems

Nano-medicine despite being regarded as an opportunity to face challenges in malaria treatment, the emerging issues like assessment of safety, biological fate, industrial chances, cost-effectiveness, and transferring the knowledge from laboratorial trials to real clinical trials still require further in-depth investigations for safe, efficient and feasible future application (*Zhang et al., 2020*).

### The combination therapy

To compete challenges in malaria treatment combination treatments is replacing single therapeutic protocols. Combination treatments involve non-artemisinins *e.g.*, sulfadoxine+ pyrimethamine, sulfadoxine + pyrimethamine + amodiaquine, and artemisinins *e.g.*,

artesunate + amodiaquine, artesunate + mefloquine, artemether + lumefantrine, artesunate + sulfadoxine/pyrimethamine (*Erhirhie et al., 2021*). Curcumin-loaded liposomes combined with $\alpha$/ $\beta$ arteether showed therapeutic efficacy against malaria compared with liposome formulation alone; in addition, recrudescence was significantly prevented (*Aditya et al., 2012*). The Artemisinin-based combination therapies for uncomplicated *Plasmodium falciparum* infection in endemic regions have been recommended by the WHO *e.g.*, Artemether-Lumefantrine (Art-L) (Coartem®) (*World Health Organization, 2015*; *Assefa et al., 2010*). However, resistance among artemisinin and its partner drug continues to evolve, producing *Plasmodium* strains more capable of surviving treatment, which can subsequently spread across a wider geographical area. As a result, triple drug combinations mefloquine + dihydroartemisinin + piperaquine and amodiaquine + artemether + lumefantrine appeared to counterbalance the resistant mechanisms of the parasite; besides being well tolerated and safe. Adverse effects were mainly in the form of loss of appetite, nausea, and vomiting in some cases. Atovaquone + pyronaridine is also recommended in combination with different mechanisms of action (*van der Pluijm et al., 2021*).

*The anti-relapse therapies.* Tafenoquine which is a long-acting 8-aminoquinoline compound belonging to primaquine has been recently introduced. In 2018, the Food and Drug Administration reported afenoquine as an anti-relapse treatment (trade name Krintafel and Arakoda) (*Haston, Hwang & Tan, 2019*).

# CONCLUSION

Further than conventional microscopy, malaria particularly in its complicated form can be mirrored through diverse altered biochemical and hematological indices *e.g.*, dyslipidemia, hypoglycemia, and elevated liver and renal functions. Of note, the aforementioned physiological anomalies are common outcomes of many metabolic and lifestyle disorders. Thus, the current review suggests comprehensive coupling of the laboratory indices with other malaria diagnosis procedures with a core emphasis on promoting the development of high-performance and highly-sensitive detection methods to refine the definitive diagnosis of malaria. In this context, we found that the exploration of biosensors is one of the promising methodologies for feasible, sensitive, and efficient malaria diagnosis. Falcipan-2 as a therapeutic target has been extensively manipulated in recent chemical therapeutic trials. The efficacy of plant extract remains a matter of research despite being derived from old folklore as having anti-plasmodial activities. Antimalaria nano-medicine demonstrates the emergence of nanocarriers (either active or passive) in drug transportation. Recent trends in antimalaria nano-medicine involve polymeric wall encapsulation of lipophilic drugs, green silver nanoparticles, and solid nanoparticles that however are under trial to increase the efficacy of antimalaria nano-medicine. The emerging drug resistance mandated trials for safe combination treatment *e.g.*, amodiaquine + artemether + lumefantrine. Also, afenoquine has been introduced as a new anti-relapse therapy.

### Study limitation

Due to human ethical considerations, the majority of studies conducted on malaria were carried out on unnatural host-parasite interaction *e.g.*, murine models that are affected by factors related to differences in host specificity and physiological functions. Thus, caution should be applied when translating those findings on human pathophysiology bearing in mind the discrepancy in metabolism and/or physiological differences among species.

### Funding

The work was supported by the Deanship of Scientific Research at King Khalid University through large group Research Project under grant number RGP2/302/44. The funders had no role in study design, data collection and analysis, decision to publish, or preparation of the manuscript.

### Grant Disclosures

The following grant information was disclosed by the authors:
Deanship of Scientific Research at King Khalid University: RGP2/302/44.

### Competing Interests

The authors declare there are no competing interests.

### Author Contributions

- Enas El Saftawy conceived and designed the experiments, performed the experiments, analyzed the data, prepared figures and/or tables, authored or reviewed drafts of the article, and approved the final draft.
- Mohamed F. Farag conceived and designed the experiments, performed the experiments, analyzed the data, prepared figures and/or tables, authored or reviewed drafts of the article, and approved the final draft.
- Hossam H. Gebreil conceived and designed the experiments, performed the experiments, analyzed the data, prepared figures and/or tables, authored or reviewed drafts of the article, and approved the final draft.
- Mohamed Abdelfatah conceived and designed the experiments, performed the experiments, analyzed the data, prepared figures and/or tables, authored or reviewed drafts of the article, and approved the final draft.
- Basma Emad Aboulhoda conceived and designed the experiments, performed the experiments, analyzed the data, prepared figures and/or tables, authored or reviewed drafts of the article, and approved the final draft.
- Mansour Alghamdi conceived and designed the experiments, performed the experiments, analyzed the data, prepared figures and/or tables, authored or reviewed drafts of the article, and approved the final draft.
- Emad A. Albadawi conceived and designed the experiments, performed the experiments, analyzed the data, prepared figures and/or tables, authored or reviewed drafts of the article, and approved the final draft.

- Marwa Ali Abd Elkhalek conceived and designed the experiments, performed the experiments, analyzed the data, prepared figures and/or tables, authored or reviewed drafts of the article, and approved the final draft.

## Data Availability

This is a literature review.

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
