# Peer review of "Malaria: biochemical, physiological, diagnostic, and therapeutic updates"

_PeerJ, doi:10.7717/peerj.17084_

## Round 0.1 · original submission · Major Revisions

I have completed my evaluation of your manuscript. The reviewers recommend reconsideration of your manuscript following major revision. I invite you to resubmit your manuscript after addressing the comments below. When revising your manuscript, please consider all issues mentioned in the reviewers' comments carefully: please outline every change made in response to their comments and provide suitable rebuttals for any comments not addressed. Please note that your revised submission may need to be re-reviewed.

Reviewer 1 ·

Basic reporting

In the article Malaria: Biochemical and Physiological Insights and the Diagnostic and Therapeutic Updates" the author attempts to review the knowledge gap and update the status of malaria diagnosis strategies. This is an interesting review recognizing the pathophysiology indices, a key knowledge that could potentiate the diagnosis and help in designing new points of care malaria diagnosis tools by using biochemical and hematological alterations. The authors also recommend future implications of Nanomedicine and plant extracts for malaria control could be important.

Experimental design

This article is a review, and authors included an extensive literature search strategy.

Validity of the findings

The authors made great efforts to refine and update the status, as well as proposed hypotheses for improving malaria diagnosis strategies. However, this reviewer feels a knowledge gap does exist in certain aspects of the study could be revised in light of the comments mentioned in the next section i.e additional comments.

Additional comments

Review comments

Limitation: The author must mention some limitation points in the review such as:
Most of the studies were carried out on unnatural host-parasite interaction; the change in metabolism or physiological changes may not be translated to human pathophysiology.


Hypoglycemia induced by malarial infection

Line 97-98: In summary, one or more of the following can also contribute to hypoglycemia in malaria:

The author mentioned that malaria-related loss of appetite contributes to the downregulation of AMP-activated protein kinase (AMPK) gene expression, and reduced energy intake.

How does AMP-activated protein kinase (AMPK) gene expression induce hypoglycemia (reduction in glucose level ) in hosts? in the mentioned ref(20) AMPK affects lipid metabolism and reduces energy intake in the context of lipid metabolism rather than glucose.


Dyslipidemia induced by malarial infection

Whether the decrease in lipids in human serum is utilized by parasites or is altered pathophysiology due to the release of proinflammatory cytokine. Please elaborate this in a few lines with the latest references.


Oxidative stress

There are several factors attributing to oxidative stress in malaria such as the release of heme from RBCs, augmented level of oxidative enzyme, and release of ROS and RNS from immune cells to invaded parasites; it is unclear which one factor is dominantly accountable to oxidative stress in malaria cases. The author mentioned only one factor.
Please elaborate on the various other factors accounting for OS in malaria cases in brief.




RDT

Highlight different variants/types of RDT

Suggest the limitation of conventionally used RDT such as a decrease in peripheral parasitemia owing to plasmodium sequestration


Therapeutics

Briefly explain under subheading disadvantages such as efficiency, bioavailability, pharmacokinetics of using conventional drugs and their side effects, etc.


Also, highlight different nano-mediated drug delivery systems in tabulated form and elaborate on some limitations of nano drugs in malaria.



Conclusion segment:

The author advocates focusing on laboratory indices to push definite diagnoses in highly suspicious cases. However, physiological anomalies such as hypoglycemia, dyslipidemia, elevated liver enzymes, and renal dysfunctions are common outcomes of many metabolic and lifestyle disorders. Thus author need to define specifically how it is relevant to malaria diagnosis, or does the author promotes coupling the laboratory indices with other malaria diagnosis process with a core emphasis on developing high-performance and highly sensitive detection method?

Reviewer 2 ·

Basic reporting

This is good study but the followings are recommended to be modified.
Introduction:
 In the first paragraph, please use the 2023 World Malaria Report data.
Result and Discussion:
 Under 3.2. Malaria Diagnostics techniques, it is better if you add the drawback of RDTs (including PfHRP2 antigen cannot be useful in Africa, India, Asia, middle east and Amazon regions since some isolates lacking this antigen have been found from these areas) and the necessity of other diagnostic techniques.
 Under 3.2. Malaria Diagnostics techniques, please replace “In 2017, Africa was reported as the main consumer of the test and consumed up to 80% of the entire RDT sales” with current information.
 Under 3.2.3 The Lab-on-a-chip and micro-device technology, you described multiplex PCR but what if you add about a species-specific loopmediated isothermal amplification (LAMP) PCR method and other recent PCR techniques.
 It is better if you add something about the rolling circle enhanced enzyme activity detection (REEAD) and micromagnetic resonance reflaxometric (MMR) tests.

Experimental design

No comments

Validity of the findings

no comments

Additional comments

Detailed correction of the paper regarding the topographical, grammatical and other errors are provided through the track changes attached herewith.

Annotated reviews are not available for download in order to protect the identity of reviewers who chose to remain anonymous.

Reviewer 3 ·

Basic reporting

The article effectively compiles data on recent diagnostic technologies and therapeutic approaches, including chemical and plant-based treatments, and emerging nanomedicine applications giving very broad and cross-disciplinary interest. It also discusses the biochemical and physiological alterations in malaria, providing a broad perspective on the disease's impact on human health. Regarding the causes, detection, and treatment of malaria, there are many similar review articles in the database, so the novelty of this article should be emphasized more. For example, it might focus on the latest advancements in nanomedicine for malaria treatment, which could be a relatively newer area in previous reviews.

While the paper is generally clear, there may be opportunities to enhance clarity and readability further, particularly in complex sections discussing biochemical and physiological insights. There were also some minor problems of wording, for example:

Line 34: The phrase "grieve morbidity". the author may be referring to "grave morbidity" to correctly describe the serious nature of the disease.

Line 87: "Peer-reviewed research, systematic and narrative reviews, and case reports that were published and met the inclusion criteria". the authors may refer to "are" instead of "and"

"Nano medicine" or "nanomedicine," please keep the wording consistency.

Line 385: The term "tubovesicular membrane (TVM)" is introduced abruptly. A brief explanation or definition of TVM when it is first mentioned would enhance understanding

Experimental design

The article's utilization of Google Scholar, PubMed, Web of Science, and the Egyptian Knowledge Bank as sources for article gathering is noteworthy, yet it falls short in offering a comprehensive and transparent methodology for article selection, which is crucial for ensuring a bias-free and all-encompassing literature review. A more robust approach would involve explicitly detailing the selection criteria for these articles, including the exact search terms used across each database. Specific keywords, especially those relating to various aspects of malaria such as "malaria diagnosis," "malaria treatment," and "biochemical changes in malaria," should be clearly stated, along with criteria used to filter studies based on their recency and relevance.

Also, this paper would significantly benefit from an elaboration on the search strategies employed in each database, including the use of filters like date ranges, article types, and language preferences, as well as any database-specific search techniques. Terms such as "current" and "older studies" need clarification to precisely convey the timeframe of the research considered. Additionally, a more meticulous approach to including or excluding articles is necessary. This should encompass a rigorous screening process of titles and abstracts for relevance, an assessment of the quality of studies, and the exclusion of certain types of articles, like case reports or editorials, which may not meet specified methodological standards. The inclusion criteria should also consider the type of study, such as clinical trials or observational studies, and the study's focus, particularly those addressing specific aspects of malaria to enhance the credibility and specificity of the review.

Validity of the findings

while commendable for summarizing the strengths and weaknesses of various malaria treatment methods, This paper could have contained more depth in its analysis to provide critical judgment and comparative insights. The authors tried to overview "the recent therapeutic trends of malaria" A crucial addition would be direct comparisons between different malaria therapeutic methods. This should encompass a detailed examination of their efficacy, side effects, cost-effectiveness, and ease of administration, among other pertinent factors.

Beyond simply summarizing the strengths and weaknesses, a more nuanced discussion is required, delving into why certain treatments might be more effective in specific contexts or how resistance issues affect their efficiency more qualitatively. It would be better to critically evaluate the research gaps and limitations in current studies on malaria treatments. This entails a thorough discussion of the constraints of existing research, such as limited sample sizes, brief follow-up periods, and homogeneity in study populations.

It may be great to provide evidence-based judgments about the reliability and future potential of different therapeutic, underpinned by data from the reviewed studies. The discussion should also extend to the public health implications of these treatment methods, including considerations of accessibility, and scalability. it would be good to identify promising areas of future research, highlighting treatment methods that exhibit significant potential for further study. Additionally, While it mentions several new technologies and therapeutic methods, such as nanomedicine and advanced diagnostic tools, it falls short of identifying specific unresolved questions or gaps in the current research, an exploration of how emerging technologies, such as nanotechnology, could revolutionize malaria treatment would provide valuable forward-looking insights. It could also highlight areas where further research is critically needed. By providing more specific suggestions for future research, the paper would not only guide ongoing studies but also stimulate new lines of inquiry in malaria research.

Additional comments

The figures in this paper are a bit blurry. Please consider replacing them with clearer ones

Reviewer 4 ·

Basic reporting

The title has grammatical errors that need to be corrected. An example can be “ Malaria: Biochemical, physiological, diagnostic and therapeutic updates”.

Line 72 the unfavorable physicochemical side effects corrects the spelling of Physiochemical.
Line no 145,  heading Liver dysfunction:

It would be great if the authors discussed the molecular mechanism of liver dysfunction during malaria. Previous studies suggest the involvement of hepatic free heme overload, nf-κb activation, and neutrophil infiltration as a major cause of liver dysfunction

Line no161: Oxidative stress: The author only discussed the host-induced oxidative stress here. However, it is very important to note that the malaria parasite resides in the highly oxygenated environment (the host RBCs) where it generates a vast amount of toxic free heme which also imposes oxidative stress on the parasite. Management of oxidative stress in malaria parasites is tightly regulated through active redox and antioxidant defense systems. The malaria parasite utilizes several important mechanisms to deal with this including the use of heme detoxificationprotein to convert free heme to hemozoin - authors need to readdress this section properly with appropriate references.

Experimental design

No comment

Validity of the findings

No comment

Additional comments

No comment

---

## Round 0.2 · accepted · Accept

It is a pleasure to accept your manuscript entitled " Malaria: Biochemical, physiological, diagnostic, and therapeutic updates" in its current form for publication in PeerJ.

Reviewer 3 ·

Basic reporting

The updated manuscript has met my suggested changes and made good improvements in terms of scientific rigor, depth, and relevance. They have improved their wording and enhanced the clarity and robustness of their methodology, and their detailed account of the article selection process and inclusion criteria has increased the credibility of the research approach. The treatment methods include evaluations and identify promising future research areas. I think it gives a good depth to the study and is in line with current trends in malaria research. The comprehensive view provided will be valuable to both researchers and practitioners.

they gave a considerable amount of future research possibilities, in terms of the dynamic nature of malaria studies and the ongoing need for innovative methods

Experimental design

no comment

Validity of the findings

no comment

Additional comments

no comment

Reviewer 4 ·

Basic reporting

The authors fulfill all the comments and justified in the text. The manuscript is perfectly suitable for publication in Peer J.

Experimental design

The authors fulfill all the comments and justified in the text. The manuscript is perfectly suitable for publication in Peer J.

Validity of the findings

The authors fulfill all the comments and justified in the text. The manuscript is perfectly suitable for publication in Peer J.

Additional comments

The authors fulfill all the comments and justified in the text. The manuscript is perfectly suitable for publication in Peer J.